# Objective Measures to Assess Active Commuting Physical Activity to School in Young People: A Systematic Review Protocol and Practical Considerations

**DOI:** 10.3390/ijerph17165936

**Published:** 2020-08-15

**Authors:** Pablo Campos-Garzón, Javier Sevil-Serrano, Yaira Barranco-Ruíz, Palma Chillón

**Affiliations:** 1Department of Physical Education and Sports, PROFITH “PROmoting FITness and Health through Physical Activity” Research Group, Sport and Health University Research Institute (iMUDS), Faculty of Sport Sciences, University of Granada, 18071 Granada, Spain; jsevil@unizar.es (J.S.-S.); ybarranco@ugr.es (Y.B.-R.); pchillon@ugr.es (P.C.); 2Department of Didactics of the Musical, Faculty of Health and Sport Sciences, Plastic and Corporal Expression, University of Zaragoza, 22001 Huesca, Spain; 3Department of Physical and Sports Education, Faculty of Education and Sport Sciences, University of Granada, 52071 Melilla, Spain

**Keywords:** physical activity, exercise, cycling, walking, young, transport, school

## Abstract

There are no systematic reviews that have identified the existing studies assessing active commuting physical activity (PA) to and from (to/from) school using objective measures, as well as the contribution of both walking and cycling to/from school to PA levels. To fill this gap in the literature, this systematic review will aim (a) to identify existing studies that assess active commuting PA to/from school with objective measures in young people and to examine the contribution of walking and cycling to/from school to PA levels, and (b) to propose an appropriate methodology and practical considerations to assess active commuting PA to/from school based on the studies identified. The review protocol was registered in PROSPERO (CRD42020162004). We will conduct a systematic search up to 2020 in five databases: PubMed, Web of Science, SPORTdiscuss, Cochrane Library, and National Transportation Library. Both the risk of bias and the quality of the identified studies will be evaluated through different instruments according to the design of each study. This systematic review will help to choose the most appropriate objective measures to assess active commuting PA to/from school and to promote walking and cycling to/from school to increase PA levels.

## 1. Introduction

Regular physical activity (PA) provides numerous physical, psychological, and social benefits in adolescents [1]. To achieve these health benefits, young people should accumulate at least 60 min of daily moderate-to-vigorous physical activity (MVPA), according the recommendations provided by the World Health Organisation and other international organisations for children and adolescents between 5 and 17 years [2]. Despite the well-known benefits of PA, more than 80% of the youth population do not meet these PA recommendations [3].

Active commuting to and from (to/from) school (ACS) by walking or cycling represents an opportunity to increase daily PA levels in children and adolescents [4,5]. According to a previous systematic review with meta-analysis conducted by Martin et al. [5], the contribution of walking to school to MVPA on school days ranges from 13 min/day in secondary-school students to 17 min/day in primary-school students. ACS not only increases PA levels but also provides other health, environmental, and economic benefits [4,6,7]. However, despite the well-documented benefits of ACS, the percentage of children and adolescents who actively commute to school has been drastically reduced in the last years in some countries, such as Australia [8], United States [9], Spain [10], New Zealand [11], Canada [12], United Kingdom [13], or Vietnam [14].

The prevalence of ACS has been mainly assessed by self-reported measures, chiefly questionnaires [15]. Although self-reported measures may be useful to identify the mode of commuting (i.e., walking, bicycling, car, motorbike, bus, train, or other), they do not allow objectively assessing the contribution of ACS (i.e., walking and/or cycling) to light or MVPA daily levels. The use of self-reported measures can be justified in large epidemiological studies. However, for a better understanding of PA, it has been suggested the need to evaluate both self-reported and objective measures [16,17].

It is important to highlight that there is no consensus on which is the best device and which methodology is appropriate to measure PA during the route to/from school. For example, Kek et al. [18] assessed the commuting-related PA through accelerometers, however, Gao et al. [19] used pedometers. There are different advantages and disadvantages to using these objective measures to assess active commuting PA to/from school. For example, although accelerometers may provide accurate information regarding frequency, intensity, and the amount of time spent in PA, depending on the location [20], these devices do not record the activity while participants are swimming and cycling. However, although heart rate monitors do not have all the advantages of accelerometers, they may be useful to capture exercise intensity in cycling or slow walking [21].

In the last years, the improvement of objective measures, and their combination has facilitated the evaluation of PA in free-living conditions [22]. Combining several objective measures has also allowed capturing more diverse information including intensity, number of meters, time, etc. [23]. For instance, several authors such as Pizarro et al. [24] and Villa-Gonzalez et al. [25] assessed the commuting-related PA to/from school through both accelerometer and Global Positioning System (GPS). Several authors have suggested that the combination of data from these two objective measures (i.e., accelerometers and GPS) might allow evaluating cycling to school properly [24,25]. Tarp et al. [20] combined GPS, accelerometers, and heart rate to assess commuting-related PA among adolescent who cycled to school. Remmers et al. [26] combined up to three objectives measures (accelerometer, GPS, and Geographic Information System [GIS]) to assess active commuting PA after school. Given the high variety of objective measures used independently and in combination, it is important to conduct a systematic review of which devices are usually used to assess walking and/or cycling PA to/from school. It will be useful to propose an appropriate methodology and practical considerations to assess active commuting PA to/from school.

There are some systematic reviews that have examined existing studies that assess active commuting PA to/from school using both self-reported and objective measures in young people, which could introduce bias in the findings [4,27,28]. Moreover, to our knowledge there is only one systematic review with meta-analysis that has the examined the contribution of walking to school to MVPA levels [5]. However, this mentioned study only assessed walking MVPA to/from school using heart rate monitoring, accelerometry, combined heart rate monitoring-accelerometry, and direct observation [5]. Therefore, light PA levels, cycling PA to/from school, and other objective measures were not taken into account [5]. In addition, given in the last year the number of studies about ACS has increased [18,19,20,24,25,26,29,30,31,32,33] and the literature search in the aforementioned review was conducted over 4 years ago, it seems necessary to update this systematic review [5]. Furthermore, only one systematic review has identified the different data collection protocols and data processing criteria of the studies that analysed PA, sedentary time, sleep time, and PA energy expenditure (PAEE) through ActiGraph GT3X, in all population (i.e., preschoolers, children, adolescents, adults, and older adults) [34]. This systematic review only focuses on one device (ActiGraph GT3X) and in total PA, without examining PA in free living conditions (e.g., ACS). Therefore, to our knowledge, this will be the first systematic review that will identify and classify the different data processing protocols and data processing criteria of the studies that analysed PA during the route to/from school using objective measures.

Therefore, the present study attempts to fill these gaps in the literature by conducting a systematic review that will provide information about the different objective measures used to assess active commuting PA to/from school in young people, and the contribution of walking and cycling to/from school to PA levels. Thus, the aims of the systematic review presented in this protocol will be (a) to identify existing studies that assess active commuting PA (total PA: e.g., counts/min, steps, PAEE, METs, kilocalories; intensity: i.e., light, moderate, vigorous, and moderate-to-vigorous) to/from school with objective measures in young people and to examine the contribution of walking and cycling to/from school to PA levels, and (b) to propose an appropriate methodology and practical considerations to assess active commuting PA to/from school based on the studies identified.

## 2. Materials and Methods

The protocol for this systematic review was registered in PROSPERO International Prospective Register of Systematic Reviews (CRD42020162004). The Preferred Reporting Items for Systematic Reviews and Meta-Analyses Protocols (PRISMA-P) guidelines [35] were used to conduct this systematic review protocol. The PRISMA-P checklist of this protocol has been included in Appendix A. The search strategy will be conducted to identify studies that assess active commuting PA to/from school with objective measures in young people. Relevant modifications of this protocol will be indicated in the corresponding systematic review.

### 2.1. Eligibility Criteria

This systematic review will include cross-sectional, longitudinal, and interventional designs (i.e., randomised trials (i.e., cluster randomised trials) and non-randomised studies (e.g., quasi-experimental studies, matched studies, non-matched studies, single group, and pilot studies)) to assess active commuting PA to/from school with objective measures in young people. In the studies with several measurement times (i.e., longitudinal and intervention studies), the information and data of the 1st measurement time (i.e., baseline) will be considered. Studies that include devices to evaluate other characteristics of the route (e.g., distance) that do not assess active commuting PA to/from school will be excluded. At least one objective measure to assess active commuting PA to/from school will be required to be included in this systematic review. The grey literature (e.g., protocol studies, reviews, editorials, and abstract or congress communications, etc.) [36] will be excluded in the search strategy. Furthermore, it is important to highlight that there is no consensus in the scientific literature about the interval time to assess commuting PA to/from school [5]. While some studies do not indicate the interval time to assess active commuting PA to/from school [37], other studies suggested different time interval ranges and number of hours. For example, Mendoza et al. [38] considered 2:30 h before school and 1:30 h after school (from 6:30 a.m. to 9:00 a.m. and from 2:30 p.m. to 4:00 p.m., respectively), and Kek [18] considered 1:00 h before and after school (from 8:00 a.m. to 9:00 a.m. and from 3:00 p.m. to 4:00 p.m., respectively). Thus, in line with the meta-analysis conducted by Martin et al. [5], all eligible studies which have not included a representative interval time for assessing active commuting PA will be excluded from the meta-analysis. Therefore, the four inclusion criteria in this systematic review will be (1) population: studies with young people (i.e., children and adolescents) who active commute to/from school in free-living conditions (from 6 to 18 years old); (2) language criterion: studies in English and Spanish, whose title and abstract will be written in English, in peer-reviewed journals; (3) topic criterion: studies analysing active commuting PA to/from school using objective measures (accelerometer, pedometer, multisensory-device, activity monitor, activity tracker, fitness information system, heart rate monitor, smartphone, APP, arm band, inclinometer, portable monitor, Fitbit, Vivofit, Fuelband, Actical, and Genea) in young people; (4) active commuting PA: studies reporting PA (total PA: e.g., counts/min, steps, PAEE, METs, kilocalories; intensity: i.e., light, moderate, vigorous, and moderate-to-vigorous) using active commuting modes (i.e., walking and/or cycling) to/from school.

### 2.2. Search Strategy

Following the recommendations of Gusenbauer & Haddaway [39] and previous systematic reviews in ACS domain [40,41], a systematic search in five electronic databases (Pubmed, Web of Science, SPORTdiscuss, Cochrane Library, and National Transportation Library) will be carried out up to 2020. A PICO strategy was performed to frame the research question and the evidence search [42]. Table 1 shows the PICO strategy (category, definition, and search terms) used for the search strategy.

The search terms were related to the following topics: young population, objective measures (e.g., accelerometer, pedometer, etc.), and active commuting PA to/from school. Search keywords were combined with different Boolean operators (i.e., “OR” and “AND”). Different search terms according to the characteristics of each database were used: Title/abstract in PubMed, topic in Web of Science, full text in SPORTdiscuss, title/abstract/key words in Cochrane Library, and Abstract in National Transportation Library. The full search strategy of this protocol has been included in the Appendix A. It has to be noted that the selected keywords are based on previous systematic reviews on this topic [1,4,5,28,34,40,41].

### 2.3. Study Selection

The study selection will be carried out by two authors (P.C.-G. and J.S.-S.) in three steps as recommended in the literature [36]. In the first step, titles and abstracts will be screened and will be chosen based on selection criteria. In the second step, full-text articles of eligible studies will be reviewed for inclusion. In the third step, the references of the selected articles will be carefully analysed to identify any other articles that could have been ignored in our search strategy. In addition, we will analyse the references of systematic reviews similar to our topic, in order to identify any other articles that meet our inclusion criteria. Discrepancies will be resolved by discussion between the two authors (P.C.-G. and J.S.-S.) or, if necessary, the other two authors (Y.B.-R. and P.Ch.) will act as mediators. EndNote citation manager software will be used to store search results as well as to remove duplicate studies.

### 2.4. Data Extraction

Four authors will form pairs for data extraction, but the main author will always be in those pairs (that it is to say, P.C.-G. will form one pair with J.S.-S., another pair with Y.B.-R., and another pair with P.Ch.). Therefore, the four authors will identify and carefully select the required information: author(s), year of publication, country, participants (sample size, mean age), study design, assessment of ACS (self-reported and/or by means of an objective device), device(s) used, objective measurement(s) methodology, device specification to evaluate MVPA, active commuting PA to/from school (walking and cycling separately if there is available data), and PA levels in schooldays. Discrepancies will be resolved by discussion between the two authors that will participate in the data extraction and, if necessary, the other two authors will act as mediators. All the extracted data will be synthesized using tables created with Microsoft Excel based on the type of objective measure.

### 2.5. Risk of Bias and Quality Assessment of the Included Studies

Following the recommendations of a recent systematic review [36], both the risk of bias and the quality of the identified studies will be evaluated. These two terms are usually confused with each other in the research literature [43]. Risk of bias refers to ‘systematic errors’ in results that may cause potential bias in the findings [44], while quality assessment refers to the degree of confidence in the results of a study [36].

As recommended by a previous systematic review [38], the risk of bias will be evaluated according to the “The Cochrane Risk of Bias Tool 2” for randomised studies [45], “ROBINS-I” for non-randomised studies [46], and an adaptation of “The Cochrane Collaboration’s Tool for Assessing Risk of Bias” for observational studies [47]. Cochrane Risk of Bias Tool 2 [45] includes 5 items: (1) bias arising from the randomisation process; (2) bias due to deviations from intended interventions; (3) bias due to missing data; (4) bias in measurement outcomes; and (5) bias in the selection of the reported result. With regard to ROBINS-I [48], this tool includes 7 items: (1) bias due to confounding; (2) bias due to selection of participants; (3) bias in the classification of interventions; and the last 4 items of the mentioned Cochrane Risk of Bias Tool 2. Finally, the Cochrane collaboration’s Tool for Assessing Risk of Bias (observational studies) [49] includes 4 items: (1) selection bias; (2) performance bias; (3) attrition; and (4) selective reporting bias. Following previous recommendations [40], studies will be classified as “high risk” if most items are rated with “some concerns” or at least one item is rated with “high risk”; studies will be classified as “some concerns” if at least one item is rated with “some concerns” and, finally, studies will be classified as “low risk” if all the items are rated as “low risk”.

For the evaluation of no intervention studies, the quality assessment will be evaluated according to the “Quality Assessment Tool for Observational Cohort and Cross-Sectional Studies” as recommended by a previous systematic review [36]. This tool includes 14 items classified in 13 domains (NIH; https://www.nhlbi.nih.gov/health-topics/study-quality-assessment-tools; [36]) as follows: (1) research question; (2) study population; (3) groups recruited from the same population; (4) sample size justification; (5) exposure assessed prior to outcome measurement; (6) sufficient timeframe to see an effect; (7) different levels of the exposure of interest; (8) exposure measures and assessment; (9) repeated exposure assessment; (10) outcome measures; (11) blinding of outcome assessors; (12) follow-up rate; and (13) statistical analyses. The 14 items are used to evaluate longitudinal studies, while 11 out of 14 items are used to evaluate cross-sectional studies (except items 7, 10, and 13) [50]. For the evaluation of intervention studies, the “Quality Assessment of Controlled Intervention Studies” will be used. This tool also includes 14 items, but classified in 11 domains (NIH; https://www.nhlbi.nih.gov/health-topics/study-quality-assessment-tools; [36]) as follows: (1) described as randomised; (2) treatment allocation-two interrelated pieces; (3) blinding; (4) similarity of groups at baseline; (5) dropout; (6) adherence; (7) avoiding other interventions; (8) outcome measure assessment; (9) power calculation; (10) prespecified outcomes; and (11) intention-to-treat analysis. Therefore, for longitudinal and intervention studies, the maximum positive score will be 14 points, while for cross-sectional studies will be 11 points. Each item will be rated as “1” when the information is reported or moderately reported, or “0” when the information is unclear or not reported. Following previous recommendations [48], longitudinal and intervention studies will be classified into “high quality” (>10 points), “medium quality” (5–9 points), and “low quality” (<4 points), while cross-sectional studies will be classified into “high quality” (>8 points), “medium quality” (4–7 points), and “low quality” (<3 points).

Following the same procedure as in the data extraction, two authors (the main author and another researcher) will evaluate the risk of bias and the quality of the studies using the different tools recommended by Gunnell et al. [36], according to the study design. Discrepancies will be resolved by discussion between the two authors, or if necessary, the other two authors will act as mediators.

### 2.6. Data Synthesis

Depending on the results, the authors will decide to conduct a meta-analysis or a narrative synthesis according to the two criteria defined by Schönbach et al. [49]. In any case, a narrative synthesis of the characteristics and the main findings of the included studies will be conducted. In both possibilities, we will summarize in a table the details of the studies and the objective measures (e.g., accelerometer, pedometer, etc.) to assess active commuting PA to/from school in young people. In addition, a second table will include the risk of bias assessment and the quality assessment, and additional information will be described in the text. Results will be provided and analysed according to the different objective measures identified. Furthermore, studies will be split in both children (6–12 years old) and adolescents (13–18 years old). If a study has a sample of both children and adolescents it will be placed in both populations [50].

## 3. Discussion

This protocol aims to present a transparent process of the methodology that will be used to carry out a systematic review of existing studies to assess active commuting PA to/from school using objective measures in young people. This review will also allow to know the contribution of PA levels being accumulated while walking or cycling to/from school in young people.

Given there are some systematic reviews that have examined existing studies that assess active commuting PA to/from school using both self-reported and objective measures in young people [4], the current systematic review will aim to fill this gap using only objectives measures. It will be expected to find a great variety of objective devices that assess active commuting PA to/from school (e.g., pedometer [19], accelerometer [18], etc.) using different methodological approaches. Although previous studies have evidenced that the combination of some objective measures to assess commuting behaviour may provide a more objective measure of PA and allow recognizing different types and domains of PA [18,37,38], there is no clear consensus regarding a “gold-standard” measurement in this field. The advantages and disadvantages of the use of each objective measure identified in this study to assess active commuting PA to/from school will be highlighted in this systematic review. Identifying feasible, valid, and reliable objective data collection methods will allow the identification of the best way to evaluate active commuting PA to/from school in young people. Proposing practical considerations regarding the objective measures to assess active commuting PA to/from school will provide information to the researchers about which kind of objective device is the most suitable to assess active commuting (i.e., walking and cycling) PA to/from school according to the design and objectives of their studies.

Finally, this review will also provide insight into the contribution of walking and cycling to school to PA levels. A previous systematic review with meta-analysis, conducted more than 4 years ago, showed that the contribution of walking to school to objectively MVPA on school days ranges from 13 min/day to 17 min/day in secondary-school students and primary-school student, respectively [5]. Given the increase of ACS studies among children and adolescents in the last years [18,19,20,24,25,26,29,30,31,32,33] it will be also important to carry out an update of this systematic review including not only walking to/from school but also cycling to/from school. Moreover, since walking is usually a light-intensity activity and provides numerous health benefits [1] it seems interesting to know the contribution of walking and cycling to/from school to light PA levels. As recently suggested [36], one of the methodological strengths of this systematic review will be the evaluation of both the risk of bias and the quality of the studies identified. Another methodological strength is the assessment of active commuting PA to/from school with objective measures in young people. Potential limitations could be anticipated such as ignoring possible databases that we are not including and, consequently, some study or studies would not be identified in the present systematic review. We also restrict our search to only studies in English or Spanish. Furthermore, grey literature will not be included in this systematic review which could have introduced some publication bias [36]. Moreover, given that there is no consensus about the interval time to assess active commuting PA to/from school, the cut-off points [1], the device’s brand, and the device position [51], the contribution of walking and cycling to PA levels could be overestimated or underestimated [5]. A future systematic review analysing the relationship between active commuting PA to/from school using objective measures and different health indicators will be a new avenue of research.

## 4. Conclusions

This systematic review will identify existing studies that assess active commuting PA to/from school with objective measures in young people, as well as the contribution of walking and cycling to/from school to PA levels. This data will be able to choose the most appropriate objective measure/s and an appropriate methodology to assess active commuting PA to/from school according to the design and objectives of their studies. Finally, the knowledge of the contribution of walking and cycling to/from school to PA levels will make people aware of the importance of ACS as a strategy to increase total daily PA levels.

## Figures and Tables

**Table 1 ijerph-17-05936-t001:** PICO strategy: category, definition, and search terms.

Category	Definition	Search Terms
Population	Young people (from 6 to 18 years old) who active commute to/from school in free-living conditions.	Child* OR adolescent* OR preadolescent* OR juven* OR teen* young* OR youth OR student OR pupil.
Intervention	Studies which using at least one device to objectively asses active commuting PA to/from school.	Objective* OR acceleromet* OR ActiGraph* OR GT3X OR activPAL* OR pedomet* OR multisensory-device OR activity monit* OR activity tracker* OR fitness information system OR heartrate monit* OR heart rate monit* OR mobile OR smartphone* OR APP OR device OR wearable monitor* OR arm band OR inclinomet* OR portable monitor* OR Fitbit OR Vivofit OR Fuelband* OR Actical OR Genea.
Comparisons	Not applicable.	Not applicable.
Outcomes	Active commuting PA to/from school.	Commut* OR transport* OR travel* OR trip OR displacement OR cycl* OR walk* OR bicycle* OR bik* OR exercise OR physical activity AND school.

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
