# Peer review of "Objective Measures to Assess Active Commuting Physical Activity to School in Young People: A Systematic Review Protocol and Practical Considerations"

_ijerph, 2020, doi:10.3390/ijerph17165936_

Round 1
Reviewer 1 Report
There is insufficient evidence to support the systematic review being proposed (revised from a previously rejected submission). Can the authors provide examples of published papers that will meet their inclusion criteria and will be able to provide the data required to answer the questions posed by the systematic review?
The authors cite a number of references as supporting evidence of an "increase of ACS studies among children and adolescents in the last years 263 [22-25,28,29]". Problematically, these references: do not contain data on active commuting to school (refs 22 & 23); do not contain data on children (refs 22 & 23) or are systematic reviews which comprise mostly self-report data.
If the systematic review aims to assess the contribution of walking and cycling to/from school to overall PA levels, there needs to be some form of self-report to nominate the transport modality used to travel to school. This is not mentioned in the inclusion criteria nor the data extraction methods.
How will the pros and cons of each objective measure be determined from each paper? What information will be systematically extracted to answer this question? Is there sufficient evidence to suggest such information is routinely reported in the articles?
Due to these concerns, the paper is deemed unsuitable for publication.
Reviewer 2 Report
The manuscript has been improved following the comments and suggestions of the reviewers. The changes met the criterias to be published in the journal.
Author Response
We truly appreciate your kind words regarding the efforts made to undertake this study. Likewise, we would like to thank you for the comments made in your review to try to improve the quality of our manuscript. We hope you will find the paper interesting and well-suited for the readership of International Journal of Environmental Research and Public Health.
This manuscript is a resubmission of an earlier submission. The following is a list of the peer review reports and author responses from that submission.
Round 1
Reviewer 1 Report
This paper describes the protocol for a systematic review examining school-aged children's active commuting to school behaviors and associations with health outcomes. Despite systematic reviews having been published focusing on the effectiveness of active transport interventions or the association between active commuting behaviors and specific health outcomes; there appears to be no published systematic review focusing on objectively-measured active transport and a variety of health outcomes. However it is also unclear whether such studies exist, as I could not find any in the manuscript's reference list. Given that cycling is a predominant school active transport behavior and is poorly measured using objective accelerometers, is it possible that the results of a systematic review assessing school active transport and health behaviors would have too much bias to provide any meaningful conclusions? Studies using GPS measures would be able to measure cycling more accurately, but the introduction provides little detail about GPS - focusing predominantly on accelerometry. Overall, a rationale for focusing solely on objective measures hasn't been clearly established in the introduction.
The systematic review aims appear to be two-fold: examining relationships between commuting behaviors to/from school with health outcomes; and analyzing strengths and limitations of objective measures to propose practical considerations. I don't think the systematic review can do both as studies are unlikely to report extractable data for both of these elements. The review could focus instead on extracting data on health outcomes but provide a general discussion of strengths and limitations of objective measures within the Discussion section.
I recommend considering a stricter definition of health indicators in the eligibility criteria. See the (Poitras et al., 2016) paper cited in the manuscript for guidance here. Also, could the topic criterion be tightend from "analysing young's (sic) commuting behaviour to/from school using objective measures" to "provide quantification of young people's commuting behaviour to/from school using objective measures"?
Heart rate (HR) as an objective measure of physical activity was discussed in line 63, yet doesn't appear within the search terms. Will studies that measure physical activity during active commuting using HR be included?
The strength of active commuting/health indicator associations may vary across different types objective measures, particularly as accelerometry will be a poor measure of cycling. If #s are sufficient, will the effect of type of objective measure on the strength of observed associations be assessed?
Overall, the manuscript needs English-language editing.
At present, the manuscript is not suitable for publication but the systematic review protocol could be revised by focusing only on health indicators and tightening up the search strategy and eligibility criteria.
Reviewer 2 Report
Protocol is well written and results can be very interesting for the audience. As you obtain registration number for Prospero, you should add in to your manuscript. Systematic review can be published without previously publish the protocol.
Think about to add subjective methods too.
What about the differences between ages, children and adolescents? maybe you need to slip your results according to these two groups.